# More Load, Less Harm? Perceived Harmfulness of Daily Activities and Low Back Pain Beliefs in Weightlifters and Powerlifters

Josce Syrett [1], David W. Evans [2,*] and Bernard X. W. Liew [1]

1   School of Sport, Rehabilitation and Exercise Sciences, University of Essex, Colchester CO4 3SQ, UK
2   Centre of Precision Rehabilitation for Spinal Pain, School of Sport, Exercise and Rehabilitation Sciences, University of Birmingham, Edgbaston, Birmingham B15 2TT, UK
*   Correspondence: d.w.evans@bham.ac.uk

**Abstract:** The purpose of this study was to understand how weightlifting/powerlifting (WL/PL) influences low back pain (LBP) beliefs and pain-related fear, and the potential influence of training, individual, and injury characteristics on these psychological features. Responses to the Photographic Series of Daily Activities-Short Electronic Version (PHODA-SeV) and the Back Pain and Attitudes Questionnaire (Back-PAQ) were collected from 67 participants who train on WL/PL. Relevant statistics were conducted to (1) compare questionnaire scores to previously published values from the general population, (2) compare male versus female WL/PL participants, (3) estimate the correlation between PHODA-SeV and Back-PAQ, and (4) identify the most important associative factors of both scores. Only the Back-PAQ was significantly lower than the published score of 113 ($p < 0.001$). Male participants had a significantly lower PHODA-SeV score compared to female participants ($p = 0.008$), but no difference was observed for the Back-PAQ. Back-PAQ and PHODA-SeV scores were moderately correlated with each other ($r = 0.54$). One of the most important association factors was back squat weight for both PHODA-SeV ($p < 0.001$) and Back-PAQ ($p = 0.006$). Future studies are required to investigate whether frequent WL/PL training improves pain-related fear and beliefs and reduces the risk of LBP occurrence.

**Keywords:** low back pain; pain-related fear; weightlifting; powerlifting

## 1. Introduction

Weightlifting (WL) and Powerlifting (PL) are popular sports involving maximal strength performed in a single repetition [1]. The popularity of these sports has grown exponentially over the past decades, with PL participation increasing by 200% and WL by 70%, largely attributed to the rise in CrossFit popularity [2–4]. Research has suggested that the risk of injury and low back pain (LBP) may increase by using heavy weights through extreme joint positions, which both WL/PL involve [5,6]. Within WL/PL, it is suggested that spinal compression forces in elite lifters can exceed 17,000 N and 8700 N, respectively [5,6]. With The National Institute for Occupational Safety and Health (NIOSH) recommending maximal compression to the spine as 3600 N for frequent exposure [7], it may be unsurprising that LBP is one of the most prevalent injuries in WL/PL, with injury rates of 2.4 to 3.3 injuries/1000 h and 1.0 to 4.4 injuries/1000 h of training, respectively [1]. With frequent exposure to high spinal loads, it is possible that athletes training and competing within WL and PL have altered views towards spinal health [8], such as having lower fear of movement and more positive beliefs towards LBP compared to the general population.

Pain-related fear is thought to be one of the most important predictors of the onset of LBP and its symptom persistence [9,10]. To understand the relationship between fear and persistent pain, the fear-avoidance model (FAM) was developed by Vlaeyen [11].

The FAM suggests that fear may result in avoidance behaviour, depressive symptoms, reduced function, ultimately driving the persistence of pain [9,11]. The FAM has also been investigated in pain-free individuals, where it is a risk factor for the onset of LBP [12–14]. Excessive levels of fear, negative beliefs and attitudes about LBP, embodied when pain-free, could exacerbate such cognition and emotion during a painful episode. This could lead to the adoption of unhelpful behaviors (e.g., avoiding certain activities), thereby increasing the risk of disability. Interestingly, although fear and pain beliefs have been investigated in a pain-free general population, this has yet to be extended to WL/PL athletes who regularly expose themselves to postures and maneuvers involving not only high loads but also a high degree of spinal flexion. A meta-analysis of quantitative sensory testing studies reported that athletes have an increased pain tolerance and thresholds compared to non-athletes [15,16]. This suggests that frequent exposure to high-load movements in WL and PL athletes may reduce fear and alter beliefs. It is likely that frequent WL/PL training (and in the process, repeatedly exposing subjects to potentially fear-provoking encounters) may result in the desensitisation of fear to strenuous spinal maneuvers, resulting in less fear and more positive beliefs compared to the general population [17].

There is evidence for the existence of sex differences in pain-related fear [18] and pain tolerance and thresholds [19], although this has not been investigated within a cohort of athletes trained in WL/PL. It may be that, because male WL/PL athletes display greater relative and absolute strength than female WL/PL athletes [20,21], the former are exposed to more frequent high-load movements than the latter. Hence, male athletes training in WL/PL could have lower fear compared to females training in WL/PL due to desensitisation.

*Aims and Hypotheses*

To quantify pain-related fear and LBP beliefs, the Photographic Series of Daily Activities Short Electronic Version (PHODA-SeV) [22], and the Back Pain Attitudes Questionnaire (Back-PAQ) [23] were used. The present study aims to understand how WL and PL influence LBP beliefs and pain-related fear, and the potential influence of training, individual, and injury characteristics on these psychological features. It is hypothesised that Back-PAQ and PHODA-SeV scores will be lower when compared to published scores from a pain-free general population [24,25]. Second, we hypothesized that female athletes would have higher Back-PAQ and PHODA-SeV scores than male athletes. Third, it is hypothesised that Back-PAQ and PHODA-SeV scores will be positively correlated. Lastly, we hypothesised that training history (number of years trained) will have a significant association with Back-PAQ and PHODA-SeV scores.

## 2. Methods

### 2.1. Study Design

This was a cross-sectional study design and the format of this study followed that of the STROBE guidelines [26].

### 2.2. Participants

Seventy participants were recruited between October 2021 and February 2022 via social media, word of mouth, printed advertisements, and contact with Weightlifting and Powerlifting gyms across the United Kingdom. Ethical approval was received from the University of Essex Human Ethics Committee (ETH2122-0110) and participants provided written informed consent before study enrolment. Participants were eligible to enrol in this study if they met the following inclusion criteria: (i) between 18–60 years old, (ii) trained or competed in WL/PL. Participants were excluded from the study for the following criteria: (i) unable to read English, (ii) suffering from chronic LBP (pain lasting more than 12 weeks) (iii) having any active medical condition which may interfere with the participation in this study.

### 2.3. Sample Size

We based our sample size calculation on the assumption that the between-subject effect size (WL/PL vs. pain-free) would be comparable to the within-subject effect size after graded activity treatment. A previous study reported an improvement in the aggregate PHODA-SeV score after graded activity treatment by 10/100 [22]. Based on a standard deviation (SD) of 12/100 [24], this improvement would translate to an effect size of 0.8. Hence, to detect a more conservative moderate effect size of 0.5, 64 participants were required to provide a statistical power of 0.8, at an alpha of 0.05. To account for dropouts, 70 participants were recruited. Three participants were excluded from analysis due to the incompletion of the Back-PAQ and PHODA-SeV questionnaires. Sixty-seven participants were included in all subsequent analysis, which still met the planned statistical power.

### 2.4. Data Collection

All data were collected using an online survey platform, Qualtrics (Qualtrics, Provo, UT, USA). Demographic characteristics and physical training data were collected, which included sex, age (yrs), height (m), body mass (kg), sport trained (WL, PL, both), and competed (none, WL, PL, both), number of years trained in WL/PL, frequency of training in WL/PL (days/week), duration of each training session (h), and self-reported one repetition maximum (1 RM) back squat and deadlift load (kg).

The modified OSLO Sports Trauma Research Centre Overuse Injury Questionnaire was used to determine the impact of low back problems, if any, on their current training over the past week [27]. The modified OSLO is a 4-item questionnaire scored on either a 4 or 5-point Likert Scale and has high internal consistency, with a Cronbach's alpha of 0.91 [27]. Current LBP intensity over the last week was also quantified using a 0–100 visual analogue scale (0 = no pain, 100 = maximal pain).

The PHODA-SeV is used to determine the perceived harmfulness of daily activities in people with LBP but has been extended into the general population [22]. The PHODA-SeV scale presents participants with 40 images, and they are asked to rate the movement they are presented with on a scale of 0–100, with 0 being 'not harmful at all' and 100 being 'extremely harmful' [22] (see appendix for PHODA-SeV). A mean score is determined by combining all question scores, with the pain-free general population registering a mean score of $19.2 \pm 12.2$ [24]. The internal consistency of the PHODA-SeV total score, indicated by Cronbach's alpha, has been reported at 0.98, and corrected item correlations have been reported as 0.42–0.82 [28]. The concurrent validity of the PHODA-SeV against the Fear-Avoidance Belief Questionnaire has been reported as Pearson's $r = 0.25$–0.50 [29].

Two questionnaires were used to quantify back pain beliefs and pain-related fear. The Back-PAQ has been suggested as the only LBP questionnaire to focus solely on the beliefs of LBP within both pain-free and chronic LBP populations [23]. The Back-PAQ has 34 question items, each scored using a 5-point Likert scale, with scores for each item being combined to provide an overall score. Eleven items (1, 2, 3, 15, 16, 17, 27, 28, 29, 30, 31) are reverse-scored compared with the other items. Overall scores range from 34 to 170 with a higher score representing more unhelpful beliefs regarding LBP [23]. Within the general population suffering from LBP a mean score of $113.2 \pm 10.6$ has been reported [25], and the internal consistency and validity of the Back-PAQ reported as Cronbach's alpha = 0.70 [23] and Pearson's $r = 0.51$–0.77, respectively [30].

### 2.5. Statistical Analysis

All statistical analyses were undertaken in R software (R version 4.1.2) with statistical significance defined by a $p$-value < 0.05.

For the first hypothesis, an independent samples t-test was used to compare the PHODA-SeV and Back-PAQ scores in the present cohort against their respective published general population mean of 19 [24] and 113 [25] respectively.

For the second hypothesis, Pearson Correlation Coefficient was used to determine the association between PHODA-SeV scores and Back-PAQ scores. The strength of correlation

was classed as negligible ($|r| < 0.30$), low ($|r| = 0.31$–$0.50$), moderate ($|r| = 0.51$–$0.70$), high ($|r| = 0.71$–$0.90$) and very high ($|r| = 0.91$–$1$) [31].

For the third hypothesis, several pre-processing steps were undertaken before modelling. Multiple imputations were performed on all variables with missing values, regardless of the amount of missing data, using the Multivariate Imputation by Chained Equations method [32]. The random forest method was used for imputation. We imputed the data using a maximum number of iterations of 30 for imputation. All continuous covariates were scaled to a mean of 0 and a standard deviation of 1 before modelling. The OSLO scores were dichotomised as some categories had too few observations. For all OSLO items, the first level was taken as the reference, i.e., 0, and all other levels were considered as 1. For both outcomes, the ten included covariates were sex, the number of years trained, 1 RM back squat load (normalized to body mass, %BM), competition sport, training sport, and OSLO dichotomised scores.

Separate step-wise regression models with bootstrap resampling (b = 1000) for stability and accuracy [33] were used to determine which covariates were most strongly associated with the outcomes of PHODA-SeV and Back-PAQ scores. Selection stability is a challenge in variable selection, given that the variables selected in the study sample may not generalise to an external cohort [34]. The bootstrap stepwise regression follows these steps. First, a new dataset is generated using resampling with replacement from the original data. Second, starting from a model with all covariates included, a stepwise selection procedure was used to remove variables based on the Akaike information criterion (AIC). As some removed variables might improve the model once other covariates are removed, the procedure also allows the addition of already removed variables. The procedure proceeds and stops if neither adding nor removing variables yields an improvement, or not in the AIC. We summarized the stability selection results by counting the frequency (out of the 1000 datasets) in which each variable was selected and expressed it as a percentage.

## 3. Results

The descriptive characteristics of our 67 participants are presented in Table 1.

**Table 1.** Descriptive Statistics for Participants.

| Variable | $n$/N = 67 [1] |
|---|---|
| **Age (years)** | 27.70 (8.35) |
| Sex | |
| Male | 35/67 (52%) |
| Female | 32/67 (48%) |
| **Height (m)** | 1.71 (0.09) |
| **Body mass (kg)** | 74.26 (12.22) |
| **Years of training** | |
| <1 yr | 11/67 (16%) |
| 1–3 yrs | 28/67 (42%) |
| 3–5 yrs | 13/67 (19%) |
| >5 yrs | 15/67 (22%) |
| **Frequency of training (days/week)** | |
| 1–4 | 19/67 (28%) |
| 5 | 34/67 (51%) |
| 6–7 | 14/67 (21%) |
| **Session duration (h)** | |
| <1 h | 11/67 (16%) |
| 1–2 | 44/67 (66%) |
| >2 | 12/67 (18%) |
| **Training sport** | |
| WL only | 20/67 (30%) |
| PL only | 25/67 (37%) |
| WL + PL | 22/67 (33%) |

**Table 1.** *Cont.*

| Variable | n/N = 67 [1] |
|---|---|
| **Competition sport** | |
| None | 25/67 (37%) |
| WL | 19/67 (28%) |
| PL | 23/67 (34%) |
| **1 RM Deadlift (kg)** | 151.48 (50.52) |
| Missing | 6 |
| **1 RM Back Squat (kg)** | 128.64 (48.94) |
| Missing | 1 |
| **Back-PAQ** | 98.87 (15.82) |
| **PHODA-SeV** | 21.12 (15.07) |
| **Oslo—1** | |
| *"Have you had any difficulties participating in training and/or competition due to back problems during the past 7 days?"* | |
| Full participation without low back problems | 48/67 (72%) |
| Full participation with low back problems | 13/67 (19%) |
| Reduced participation due to low back problems | 0/67 (0%) |
| Cannot participate due to low back problems | 6/67 (9.0%) |
| **Oslo—2** | |
| *"To what extent have you modified your training and/or competition due to back problems during the past 7 days?"* | |
| No reduction | 46/67 (69%) |
| To a minor extent | 15/67 (22%) |
| To a moderate extent | 4/67 (6.0%) |
| To a major extent | 2/67 (3.0%) |
| **Oslo—3** | |
| *"To what extent have back problems affected your performance during the past 7 days?"* | |
| No effect | 50/67 (75%) |
| To a minor extent | 11/67 (16%) |
| To a moderate extent | 4/67 (6.0%) |
| To a major extent | 2/67 (3.0%) |
| **Oslo—4** | |
| *"To what extent have you experienced back pain related to your sport during the past 7 days?"* | |
| No pain | 44/67 (66%) |
| Mild pain | 16/67 (24%) |
| Moderate pain | 7/67 (10%) |
| Severe pain | 0/67 (0%) |
| **Pain intensity (0–100)** | 12.13 (16.19) |

[1] Mean (SD); n/N (%). Abbreviations: RM: repetition maximum, WL: weightlifting, PL: powerlifting, Back-PAQ: Back Pain Attitudes Questionnaire, PHODA-Sev: Photographic Series of Daily Activities Short Electronic Version.

Upon testing the first hypothesis, that Back-PAQ and PHODA-SeV scores will be lower when compared to published scores from a pain-free general population, we found that the PHODA-SeV score in the cohort of WL and PL athletes was non-significantly different from the population mean of 19 (95% CI = 17.44, 24.79, t (66) = 1.15, $p$ = 0.254 (Figure 1). Back-PAQ score in the present cohort was significantly lower than the general population mean of 113 (mean = 98.9, 95% CI = 95.0, 107.7, t (66) = −7.31, $p$ < 0.001) (Figure 1).

Upon testing the second hypothesis, that female athletes would have higher Back-PAQ and PHODA-SeV scores than male athletes, we found that male participants had a significantly lower PHODA-SeV score compared to female participants by a magnitude of 9.8 (95% CI = −16.9, −2.6, t (53) = −2.74, $p$ = 0.008) (Figure 1). The difference between male (96.5) and female (101.5) Back-PAQ scores did not reach statistical significance (95% CI = −12.7, 2.6, t (64) = −1.32, $p$ = 0.192) (Figure 1). A moderate positive correlation was seen between Back-PAQ and PHODA scores, r (65) = 0.54, and this relationship was significant ($p$ < 0.001).

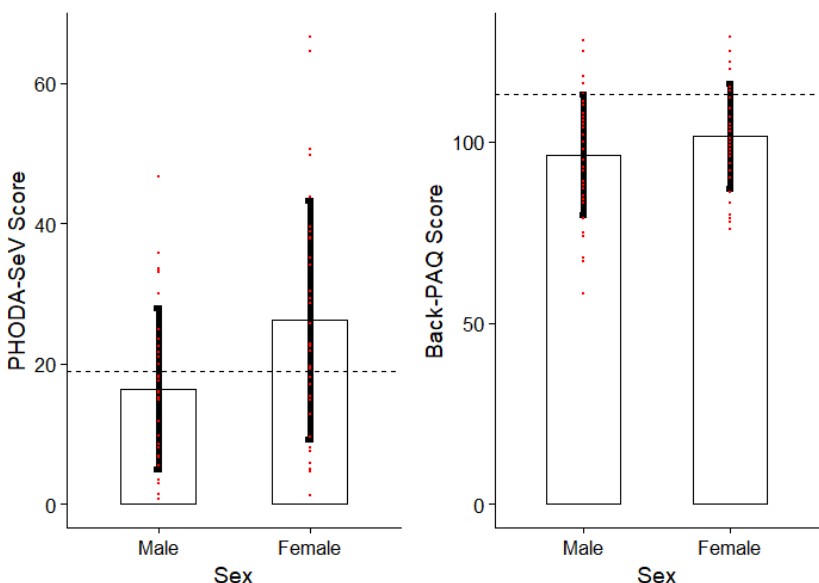

**Figure 1.** Mean ± Standard Deviation as Error Bars PHODA-SeV and Back-PAQ Scores for Males, Females. Dashed Line Represents General Population Mean Score, and red dots represent individual data points.

Our third hypothesis was that Back-PAQ and PHODA-SeV scores will be positively correlated. For the outcomes of Back-PAQ and PHODA-SeV, the final models had $R^2_{adj}$ of 0.13 and 0.37, respectively. We also hypothesized that training history (number of years trained) will have a significant association with Back-PAQ and PHODA-SeV scores. Training factors associated with PHODA-SeV were the number of years trained, and back squat weight (Table 2). For years trained, training between 1–3 yrs resulted in a 12 unit decrease in PHODA-SeV score compared to training <1 yr (Table 2). For back squat weight, a 1 SD increase in weight resulted in an 8.2 unit decrease in PHODA-SeV score (Table 2). The training factor associated with Back-PAQ was the back squat weight; a 1SD increase in weight resulted in a 5.4 unit decrease in Back-PAQ score (Table 3). For the stability analysis on the outcome of PHODA-SeV score, back squat weight and number of years trained were the two most stable covariates (Table 4). For Back-PAQ score, back squat weight and type of training sport were the two most stable covariates (Table 4).

**Table 2.** Multiple Regression Analysis for the Relationship Between Variables and PHODA-SeV Scores.

| Variable | Beta | 95% CI [1] | *p*-Value |
|---|---|---|---|
| **(Intercept)** | 28 | 20, 37 | **<0.001** |
| **Age** | −3.7 | −7.4, −0.03 | **0.048** |
| **Oslo Q2** | | | |
| 0—No reduction | — | — | |
| 1—Others | 8.5 | 0.85, 16 | **0.030** |
| **Oslo Q4** | | | |
| 0—No pain | — | — | |
| 1—Others | −5.2 | −13, 2.3 | 0.2 |
| **Years Train** | | | |
| 1—<1 y | — | — | |
| 2—1–3 y | −12 | −21, −3.5 | **0.007** |
| 3—3–5 y | −5.4 | −16, 5.7 | 0.3 |
| 4—>5 y | −8.7 | −20, 2.4 | 0.12 |
| **Back Squat Weight (kg)** | −8.2 | −12, −4.7 | **<0.001** |

[1] CI = Confidence Interval.

**Table 3.** Multiple Regression Analysis for the Relationship Between Variables and Back-PAQ Scores.

| Variable | Beta | 95% CI [1] | *p*-Value |
|---|---|---|---|
| **(Intercept)** | 98 | 94, 103 | **<0.001** |
| **Age** | −3.0 | −6.8, 0.74 | 0.11 |
| **Oslo Q1** | | | |
| 0—Full participation w/o problems | — | — | |
| 1—Others | 12 | 2.0, 22 | **0.019** |
| **Oslo Q4** | | | |
| 0—No pain | — | — | |
| 1—Others | −7.7 | −17, 1.8 | 0.11 |
| **Back Squat Weight (kg)** | −5.4 | −9.2, −1.6 | **0.006** |

[1] CI = Confidence Interval.

**Table 4.** Frequency of PHODA-SeV and Back-PAQ Covariates in Stability Analysis.

| Covariates | Phoda-SeV (%) | Back-PAQ (%) |
|---|---|---|
| Back squat weight | 98.2 | 69.1 |
| Years of training | 87.8 | 50.1 |
| Age | 75.5 | 40.3 |
| Oslo Q2 | 57.9 | 29.8 |
| Oslo Q4 | 57.4 | 52.2 |
| Training sport | 44.6 | 66.7 |
| Competition sport | 36.0 | 35.6 |
| Sex | 33.4 | 23.4 |
| Oslo Q1 | 33.3 | 56.3 |
| Oslo Q3 | 30.0 | 37.2 |

## 4. Discussion

Frequent exposure to high-load movements may reduce fear and alter beliefs in WL and PL athletes, which have not been investigated previously. Our findings did not fully support our first hypothesis that WL/PL athletes would report a lower Back-PAQ score compared to the general population, without a difference in PHODA-SeV score. Female athletes scored higher on the PHODA-SeV compared to male athletes, which although statistically significant, did not cross the minimal detectable change threshold of 20 [22]. Back-PAQ was moderately correlated with PHODA-SeV, which supported our third hypothesis. Lastly, the number of years of WL/PL training was only significantly associated with PHODA-SeV score.

WL/PL athletes have more positive beliefs compared to the general population [25] but perceive similar levels of fear [24]. A possible confounding factor to this result could be the different ages of samples in prior studies, where Christe et al. [25] included participants who were on average ~10 years older, but Knechtle et al. [24] included similarly aged participants to our study. However, Christe et al. [25] also reported that age was not significantly associated with Back-PAQ score, suggesting that our findings are resilient to age differences between studies. The diverging relationship between beliefs and fear is also reflected in our correlation analysis, which found only a moderate correlation between Back-PAQ and PHODA-SeV. In contrast, one study reported a strong correlation magnitude between Back-PAQ and the Tampa Scale of Kinesiophobia in a cohort of healthcare practitioners and the general public [35]. However, another study reported a moderate correlation between Back-PAQ and the Fear Avoidance Beliefs Questionnaire in a cohort of healthcare practitioners [30]. The divergent change to beliefs and fear suggest that other variables could moderate their relationship.

Given that the present study compared the mean questionnaire scores against the published scores from a non-matched cohort, we can only speculate as to the potential cause(s) of more positive beliefs in WL/PL athletes. It may be that frequent exposure

to high spinal-load maneuvers, performed by WL/PL athletes, improves the beliefs of specific items on the Back-PAQ, such as "Lifting without bending the knees is not safe for your back". This was supported by our regression model which found a negative association between back squat weight and Back-PAQ score. Alternatively, the trained status of the present cohort could have increased their self-efficacy [36] in managing the challenges associated with LBP, thus scoring more positively on the Back-PAQ. Another study reported that the higher the level of education status, the more positive the beliefs held by an individual [25]. This suggests that differences in the educational status between the present cohort and previous research [24,25] could contribute to the observed difference. Future research should investigate how different sports and activity statuses alter individual item scores on the Back-PAQ.

It is possible that the lack of difference in pain-related fear between WL/PL athletes and the general population could be due to the lack of movement specificity between WL/PL training and the movements depicted in the PHODA-SeV. Prior research has shown that exposure to a 'touching toes' movement to reduce fear did not reduce fear of a dissimilar straight leg raise movement [37]. The conclusion drawn from this was that patients appear to allow an exception to the rule (e.g., touching toes is safe) rather than a fundamental change to the rule (e.g., all spinal flexion movements are safe) [38]. The suggestion of movement-specificity was further supported by our results, with back squat weight having a stronger association with fear than beliefs. Researchers have identified the need for sport-specific images, with previous research showing that modification to the original PHODA-SeV questionnaire, for use with athletes recovering from knee injuries, produced a valid and reliable tool for identifying the hierarchy of fearful movements [39]. Future research comparing individual item scores on general lay activities (as depicted within the PHODA-SeV) and new sports-specific images, in the same WL/PL cohort and the general population, will be useful to understand the role of WL/PL on task-specific fear.

Females had slightly greater levels of increased pain-related fear compared to males, supporting the findings of previous research [18]. Interestingly, the effect of sex was not selected in the regression model for PHODA-SeV and Back-PAQ. By contrast, one study reported that sex was significantly associated with Back-PAQ score [25]. It is unlikely that the exclusion of sex occurred due to redundant information contained within the covariate of back squat strength. This is because we normalised strength to the individual's body mass before regression modelling, hence eliminating the component of sex differences in strength attributed to differences in body mass [20,21]. Additionally, if sex and squat weight were highly collinear, we would expect an almost equal frequency of selection in our stability selection analysis. Speculatively, greater strength might increase an individual's self-efficacy for performing activities of daily living without harm, thereby reducing their fear score on the PHODA-SeV. Alternatively, greater strength may reflect exposure to more frequent training sessions at heavier lifted loads, resulting in greater desensitisation to fear-provoking movements.

Our regression models showed that training modification (Oslo Q2) was significantly associated with greater fear, but lack of full participation (Oslo Q1) was significantly associated with more negative beliefs. A prior study in the general population reported a significant association between the presence of a current LBP episode and negative beliefs [25]. In contrast, we did not find a significant association between pain severity (Oslo Q4) and the outcomes of fear or beliefs. Our data suggest that the behavioral consequence of pain was a more important associative factor for fear and beliefs than the pain itself. It may be the case that our participants were trained to tolerate pain [40], and that having to miss or modify training was a more important stimulus for influencing fear and beliefs. However, the cross-sectional nature of the study precludes the possibility to delineate whether greater fear and more negative beliefs drive training modification and participation, or vice versa.

Our cross-sectional analysis also precludes us from concluding that training in WL/PL improves positive beliefs and reduces fear over time. A previous study reported that

high-load lifting exercises are just as effective in pain reduction, compared to low-load exercises [41]. To the authors' knowledge, effects of high-load vs. low-load training on LBP beliefs and fear in both symptomatic and asymptomatic cohorts have not been investigated. Speculatively, the present findings support the potential that frequent training in WL/PL may have a positive effect on LBP beliefs and fear, and these might reduce the risk of LBP onset.

*Limitations*

First, even though we compared our PHODA-SeV and Back-PAQ scores with published scores from the literature, we note that the comparison was made in a non-age-matched or sex-matched cohort [24,25]. Second, our cross-sectional analysis cannot disassociate between within-subject and/or between-subject associations. Longitudinal follow-ups to disentangle the causal relationship between frequent WL/PL training and the beliefs and fear about LBP would be required. Third, the present analysis of identifying the variables most associated with Back-PAQ and PHODA-SeV is exploratory and should be viewed within a hypothesis-generation framework. Future studies should confirm whether the identified variables remain the most influential variables of Back-PAQ and PHODA-SeV in an independent study.

**5. Conclusions**

Athletes who regularly trained in WL and PL reported more positive back pain beliefs, but similar levels of fear, compared to previously published scores from the general population. A greater self-reported 1 RM back squat load was associated with lower levels of fear and more positive back pain beliefs. Whether WL and PL can be used as a strategy to manage pain-related, unhelpful beliefs, and ultimately reduce the risk of LBP occurrence remains to be investigated. Future investigation into the physiological, psychological, cortical, and motor mechanisms underpinning frequent WL/PL and the development of fear and unhelpful beliefs would benefit the understanding of the link between exercise, LBP, cognitions, and emotions.

**Supplementary Materials:** The following supporting information can be downloaded at: https://www.mdpi.com/article/10.3390/app13010220/s1.

**Author Contributions:** Conceptualization, J.S., B.X.W.L. and D.W.E.; Data curation, J.S. and B.X.W.L.; Formal analysis, J.S. and B.X.W.L.; Investigation, J.S. and B.X.W.L.; Methodology, J.S., B.X.W.L. and D.W.E.; Project administration, J.S. and B.X.W.L.; Resources, J.S. and B.X.W.L.; Software, B.X.W.L. and D.W.E.; Supervision, B.X.W.L.; Validation, B.X.W.L. and D.W.E.; Visualization, B.X.W.L. and D.W.E.; Writing—original draft, J.S.; Writing—review and editing, J.S., B.X.W.L. and D.W.E. All authors have read and agreed to the published version of the manuscript.

**Funding:** This research received no external funding.

**Institutional Review Board Statement:** Ethical approval was received from the University of Essex Human Ethics Committee (ETH2122-0110).

**Informed Consent Statement:** Informed consent was obtained from all subjects involved in the study.

**Data Availability Statement:** All data and codes to reproduce the results are provided within the Supplementary Material.

**Conflicts of Interest:** The authors declare no conflict of interest.

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
