# Peer review of "More Load, Less Harm? Perceived Harmfulness of Daily Activities and Low Back Pain Beliefs in Weightlifters and Powerlifters"

_applsci, doi:10.3390/app13010220_

Round 1
Reviewer 1 Report
The work entitled “More Load, Less Harm? Perceived Harmfulness of Daily Activities and Low Back Pain Beliefs in Weightlifters and Power lifters”. Addresses a topic of great interest, with adequate and validated tools, to consider it for publication in Applied sciences. But before its publication, I would like to make the following considerations to the authors:
The abstract should be reformulated, including all sections of the work, as well as making a greater description of the development of the work.
Although the introduction perfectly focuses the work, it is supported by bibliographical references that are not very current. Of 22 references, only three references are from the last five years. I recommend authors to review these aspects.
Section 1.1, I recommend be synthesized, clearly and concisely.
I recommend that the authors reflect in the section of subjects, the dropout rate or that they reflect the final n.
The results section should be reformulated. They must write text in which the most relevant results or characteristics are exposed, and in referencing the tables for more information, if the reader so wishes. In the current text it is the reverse.
I observe absences of opinions of the authors in the discussion, as well as how these results can help to reduce the pathologies studied.
Reviewer 2 Report
Congratulations to the authors for a very well written study.
The great strength of the study is the clean methodology and statistics as well as the vey precise presentation of the results.
The greatest weakness, however, is that there is no control group with the same epidemiological characteristics. The comparison with previously published groups is legitimate, but their epidemiological characteristics should be discussed in more detail. In particular, the "age" factor should be discussed more, since the patient group is very young and the control groups are probably significantly older. For the future it would be interesting to interview older athletes.
In addition, I would recommend discussing the clinical relevance of the collected data in more detail.
Reviewer 3 Report
Dear Authors
I reviewed the topic "More Load, Less Harm? Perceived Harmfulness of Daily Activities and Low Back Pain Beliefs in Weightlifters and Powerlifters". The topic of this research is interesting and important. I think the authors must have put a lot of effort into taking an experiment and writing the manuscript. I think it will be a better thesis if only a few minor revisions presented below are revised.
Minor points:
Q1: Statistical symbol 'p' in abstract and text should be written in italics.
Q2: References need to be presented in accordance with the MDPI format.
I hope my review helped you improve your manuscript.
Best regards,
Round 2
Reviewer 1 Report
The authors made major changes to the manuscript, and this can be seen in the new version of the manuscript. However, some of the considerations, which in my opinion are of importance, were not addressed correctly, or not addressed: the bibliography was not updated, leaving a work based on non-current precedents. They did not clarify the N of the sample correctly. The objectives and results are still not clearly stated.
Author Response
Comments are in bold, response in normal typeset, and excerpts from manuscript are in italics.
Reviewer #1
The authors made major changes to the manuscript, and this can be seen in the new version of the manuscript. However, some of the considerations, which in my opinion are of importance, were not addressed correctly, or not addressed: the bibliography was not updated, leaving a work based on noncurrent precedents.
Reply: We thank the Reviewer for this question. We apologise for not addressing this in the previous round of revision. Previously, the Reviewer used a threshold of articles within the “last five years”. Some articles Cholewicki et al (1991) represent seminal research performed in rare and challenging experimental conditions (e.g. lifting extremely heavy weights) that cannot be easily replicated – and certainly have not been within the last 5 years.
We have added six more articles in the Introduction, two of which were published outside of the 5 year threshold. In our Introduction, 12 out of 25 articles are from 2017 onwards. If the Reviewer believes that there are other important references still to be cited, we would be glad to include these if the Reviewer can provide specific references.
They did not clarify the N of the sample correctly.
Reply: We thank the Reviewer for this question. We are unsure what the Reviewer meant by “clarifying the N of the sample correctly”. We have added a further sentence to read as follows (line 107):
Sixty-seven participants were included in all subsequent analysis, which still met the planned statistical power.
The objectives and results are still not clearly stated.
Reply: We thank the Reviewer for these questions. There was no mention in Revision 1 of a problem pertaining to the objectives, so we are unsure as to how we should improve this. We have explicitly stated the aims (lines 73-75) and hypotheses (lines 75-81) in the Introduction. We are very willing to reword the objectives or aims if the Reviewer (or Editor) feels this is necessary and can provide us with more guidance. We explicitly refer to these hypotheses in the methods (lines 144-161). We have now restated the hypotheses in the results section (lines 183-207) to make it easier for the reader to link these sections. We are very willing to reword the Results further if the Reviewer (or Editor) can provide further guidance.
